

# Weight loss effects of non-pharmacological interventions in women with polycystic ovary syndrome: a systematic review and network meta-analysis

Rong Hu[1], Lihong Zhang[1], Jingjing Zhu[2], Sihua Zhao[3], Lixue Yin[4] and Junping Hu[4]

[1] School of Nursing, Lanzhou University, Lanzhou, Gansu, China
[2] Medical College, Ankang University, Ankang, Shanxi, China
[3] The Second Ward of the Department of Urology, The First Hospital of Lanzhou University, Lanzhou, Gansu, China
[4] Reproductive Center, First Hospital of Lanzhou University, Lanzhou, Gansu, China

Corresponding author
Sihua Zhao, 891883958@qq.com

## ABSTRACT

**Objective:** To compare the effectiveness of non-pharmacologic interventions in improving weight loss management in overweight patients with polycystic ovary syndrome (PCOS).

**Methods:** Five databases, PubMed, Embase, Cochrane, Web of Science and China Knowledge, were searched for this study. The Cochrane risk of bias tool was used to assess the risk of bias of eligible studies. The included randomized controlled trials were subjected to traditional meta-analysis (TMA) and network meta-analysis (NMA), and the cumulative number of surfaces under the ranking curve (SUCRA) was calculated for each intervention to derive the optimal intervention.

**Results:** The study ultimately included 29 articles involving 22 different interventions and 1,565 patients. The results of the NMA showed that the optimal intervention for the four outcome measures (body weight, body mass index (BMI), waist circumference (WC), waist-hip ratio (WHR)) was nutritional supplement + low-calorie diet, Taichi, continuous aerobic exercise and Taichi.

**Conclusion:** Current evidence suggests that nutritional supplements + hypocaloric diet; Taichi; continuous aerobic exercise have the greatest clinical advantage in weight loss and deserve to be promoted in the clinic. One of the best interventions for both outcome indicators, Taichi, suggests that it may be a common misconception that simply increasing the intensity of exercise is not the only way to lose weight and improve health.

## INTRODUCTION

Polycystic ovary syndrome (PCOS) is a common endocrine-metabolic disorder characterized by anovulation, hyperandrogenism (*Rosenfield & Ehrmann, 2016*;

*Zhou et al., 2023*), and polycystic ovarian morphology, and is the leading cause of anovulatory infertility affecting approximately 5–20% of women of reproductive age worldwide (*Yu et al., 2023*; *Azziz, 2018*). In fact, PCOS does not only emphasize problems with the ovaries, but with multiple organ systems, especially as manifested in metabolic health. Patients often experience insulin resistance, weight gain, irregular menstrual cycles and hirsutism symptoms (*Taieb et al., 2024*). The etiology of PCOS is not clear, and is generally considered to be genetic, metabolic, dietary, and obesity factors, with overweight and obesity occupying a large proportion of patients with PCOS. Among individuals in China diagnosed with polycystic ovary syndrome (PCOS), 37% experience overweight as part of their health journey (*Stener-Victorin et al., 2024*). Higher body weight is not only associated with the development of PCOS but may also worsen existing metabolic and reproductive challenges in individuals diagnosed with this condition, contributing to elevated insulin resistance, increased androgen levels, and compromised ovarian function (*Cooney et al., 2017*). In addition, obesity can reduce the success of fertility assistance in individuals with infertility by increasing the incidence of anovulation and menstrual disorders, decreasing the sensitivity of clomiphene and gonadotropins to ovulation, and increasing the difficulty of treatment (*Martin, 2023*). Therefore, it is crucial for healthcare providers to support weight management strategies for individuals with PCOS who have excess body weight, as this can optimize their reproductive potential. Multiple randomized controlled trials have explored weight management strategies for individuals with PCOS and elevated body weight, incorporating both pharmacological and non-pharmacological approaches. Non-pharmacological interventions are recommended as the initial approach for weight management in individuals with PCOS, given their favorable safety profile and cost-effectiveness, particularly for those with higher body weight (*Teede et al., 2023*). However, the results of these nonpharmacological interventions in terms of effectiveness in body mass improvement were inconsistent and it was not possible to visually compare the best nonpharmacological interventions.

Network meta-analysis (NMA) can be used to analyze indirect and direct data in order to rank different interventions and finally analyze the best measures for the management of body mass in patients with PCOS (*Page et al., 2021*). This study aimed to evaluate and compare nonpharmacological strategies for weight management in individuals with PCOS through traditional meta-analysis (TMA) and, ultimately identifying optimal approaches by ranking their effectiveness.

## METHOD

### Search strategy

The research protocol was registered with PROSPERO (CRD42023458815). This study conforms to all PRISMA guidelines and reports the required information accordingly (see Supplemental Checklist).

Five databases, including PubMed, Embase, Cochrane, Web of Science, and China Knowledge, were searched for all published randomized controlled trials between the date of database creation and June 2024. The search was conducted according to the principle of subject terms combined with free text, and keywords included non-pharmacological

interventions*, polycystic ovary syndrome*, polycystic ovary syndrome, overweight, *lifestyle, *dietary therapies, *exercise therapies, cognitive-behavioral therapies, RCTs, randomized controlled trials, and so on (see Table S1 for details). Appropriate adjustments were made to the different databases and references to relevant research reviews were tracked to ensure that the complete body of qualified literature was included.

## Literature screening

### Inclusion criteria

Inclusion criteria included: (1) Inclusion in the study of patients diagnosed with PCOS according to the National Institutes of Health (NIH) Diagnostic Criteria (1990) (*Zawadri, 1992*), the ESHRE/ASRM Rotterdam (2003) Diagnostic Criteria (*ESHRE, 2004*), or the AE-PCOS Criteria (2006) Guidelines (*Azziz et al., 2006*), and the patients were diagnosed as overweight. (2) The patient only received a non-pharmacological intervention. (3) The study design was a randomized controlled trial (the control group was no treatment, placebo control, usual intervention, or other different non-pharmacological interventions). (4) Interventions are non-pharmacological and include dietary interventions, exercise interventions, nutritional supplements, cognitive-behavioral interventions, and acupuncture/electroacupuncture. Articles containing single or combinations of interventions were included. (5) The study outcome consisted of four primary outcome indicators: body mass index (BMI), weight, waist circumference (WC), and waist hip ratio (WHR).

### Exclusion criteria

Exclusion criteria included: (1) Studies that included drugs or surgery as part of the study intervention. (2) Presence of other diseases that affected the study outcome. (3) Study design of case studies, cross-sectional studies, non-randomized controlled trials, conference abstracts, and review articles. (4) Full text could not be found.

## Data extraction

After the initial screening of articles to remove duplicates through EndNote software, two researchers independently extracted data from the eligible articles and recorded it in a table. Extracted data included the author of the article, the year, the country, the age of the participants, the type of intervention group, the sample size of each study, the length of the intervention, and the outcome indicators included. Results include the mean and standard deviation of each outcome indicator at baseline and at the end of the intervention. If the original article could not be extracted directly to get the mean and standard deviation at the two time points, the mean and standard deviation were calculated using the computational public notices, and if we still could not extract what we wanted, the outcome was discarded for this article.

The blank control group received no intervention program; the usual care group was instructed to continue their current lifestyle; and subjects in the placebo group received treatments (*e.g.*, sugar pills) that were identical to those of the intervention group in terms of appearance and method of administration but "lacked active ingredients." The purpose

of this study is to examine the effects of the intervention group (a non-pharmacological intervention). By comparing the differences in the control groups across the included literature, we concluded that the variations among the control groups were negligible. Therefore, in this study, the blank control group, usual care, and placebo group were collectively referred to as the "no specific intervention group".

## Quality evaluation

Two investigators independently assessed the risk of bias for each of the included studies according to the Cochrane risk assessment tool for randomized trials. risk of bias consisted of five components: (1) Bias due to the randomization process. (2) Bias due to deviation from the intended intervention. (3) Bias due to missing outcome data. (4) Bias in the measurement of the outcome, and (5) Bias in the choice of reported outcomes. Each area was judged to be "high risk," "low risk," or "some concern" based on the content of the original article. If all five areas were judged to be low risk, the overall bias was low risk. If one area is judged to be of "some concern," the overall bias is "some concern." If one or more areas are judged to be of "some concern," then the risk of bias in the literature is high (*Sterne et al., 2019*). During the quality assessment process, if two researchers disagree (Zhang Lihong, Hu Rong), the final judgment is determined through discussion, and if the two researchers are unable to agree, the decision is made by questioning a third party (Sihua Zhao).

## Data analysis

This study used TMA using Revman software (Version 5.3) to directly compare the effects of 22 interventions such as low-fat diet, Mediterranean diet, continuous aerobic exercise and cognitive-behavioral therapy on the improvement of the main outcome indices (BMI, body weight, WC, WHR). We decided to express effect sizes in terms of mean difference (MD) for continuous data and relative risk (RR) for dichotomous data, with 95% confidence intervals (CI) and error of 0.05. If there was heterogeneity in the included studies, a random-effects model was used, and if there was no heterogeneity, a fixed-effects model was used. In this study, we categorized the data into four groups based on outcome indicators, using both the random-effects model and the fixed-effects model. If the number of included trials was greater than 10, a funnel plot was used for publication bias analysis. Funnel plots were constructed using Stata software (Version 17.1).

Next, a NMA was performed using Stata software (Version 17.1) to compare the effects of the interventions on the improvement of body mass in PCOS and to prioritize the effects of the interventions. Network diagrams depicted the comparative networks of interventions and controls. In the figure, circular nodes indicate the size of the included sample, connections indicate the presence of studies between them, and the thickness of the line indicates the number of studies. Before synthesizing the data, we evaluated the consistency assumption to check for inconsistencies.

A random effects model was used to summarize all relevant parameters. Continuous data were summarized as MD with 95% confidence intervals (CRLs), and dichotomous data were summarized as RR. The likelihood of each rank order was estimated using
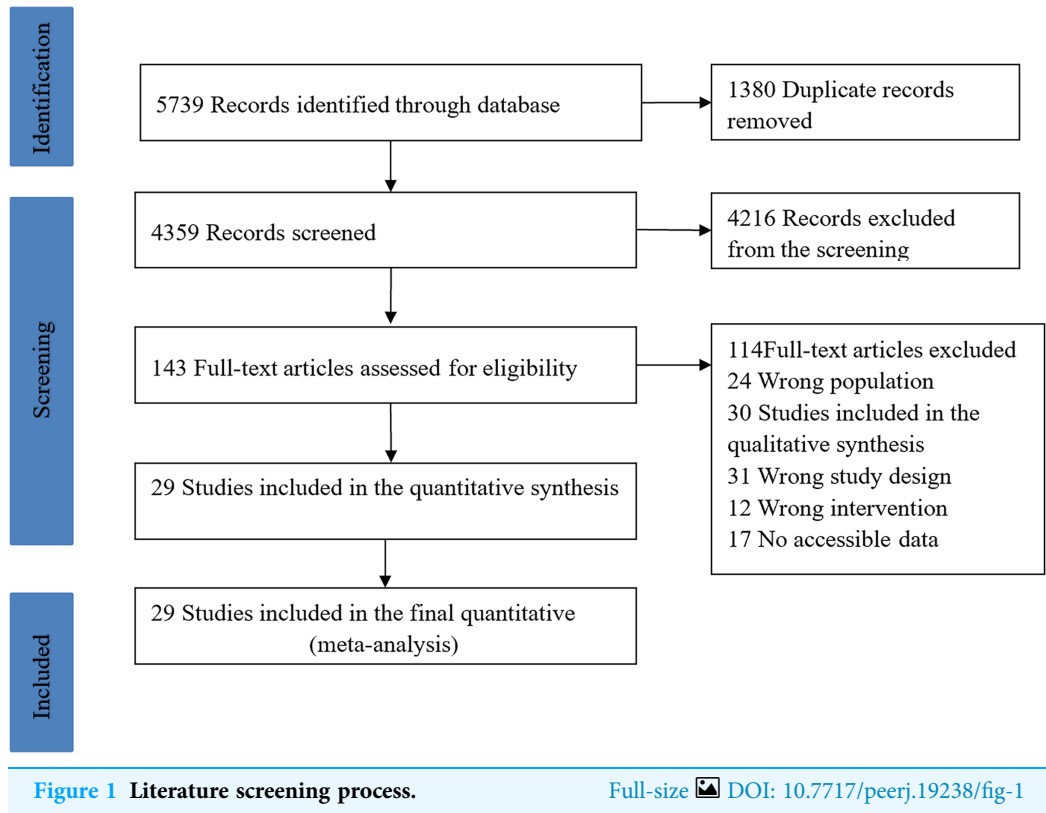

**Figure 1** **Literature screening process.**

surfaces under cumulative ranking curve analysis (SUCRA), using scores ranging from 0% to 100%. It can reveal high-value interventions, indicating a high preference for them.

## RESULTS

### Literature search and screening results

A total of 5,739 articles were retrieved and after removing duplicates, 4,359 articles were obtained. After initial screening of the 4,359 articles by title and abstract, 143 articles were obtained that met the inclusion criteria. After full-text reading and screening of the included literature, 114 articles were excluded based on the inclusion and exclusion criteria, and finally 29 articles were included in this study. The specific process of literature screening is shown in Fig. 1. The 29 included articles contained a total of 1,565 participants and 22 interventions (including one intervention alone or a combination of interventions).

### Study characteristics

Twenty-nine randomized controlled studies with a total of 1,565 subjects were included. Eleven studies were conducted in the Asia-Pacific region (China, Iran, India) and 18 studies were conducted in the non-Asia-Pacific region (Denmark, Switzerland, Canada, Australia, Italy, Norway, Turkey, *etc.*). Most studies set intervention duration at 12 weeks and beyond. Six studies contained three study groups and the others were divided into two comparison groups. Thirteen studies set the control group as no specific intervention (blank control, usual care), and the rest compared two different interventions, including

**Table 1  The characteristics of included studies.**

| Author, year | Country | Lasting days (weeks) | Study type | Sample | Intervention | | Control |
|---|---|---|---|---|---|---|---|
| | | | | | T1 | T2 | |
| Tabrizi et al. (2020) | Iran | 12 | RCT | 44 | Nutritional supplements + Hypocaloric | | Hypocaloric |
| Mei et al. (2022) | China | 12 | RCT | 59 | Mediterranean diet | | Low fat diet |
| Stener-Victorin et al. (2009) | Sweden | 16 | RCT | 20 | Acupuncture | Continuous aerobic exercise | No special intervention group |
| Thomson et al. (2008) | Australian | 20 | RCT | 52 | High-protein + Continuous aerobic exercise | High-protein diet + Continuous aerobic exercise + Resistance training | High-protein diet |
| Łagowska & Drzymała-Czyż (2022) | Poland | 20 | RCT | 40 | Nutritional supplements | | Hypocaloric |
| Ribeiro et al. (2021) | Brazil | 16 | RCT | 87 | Continuous aerobic exercise | Intermittent aerobic exercise | No special intervention group |
| Turan et al. (2015) | Turkey | 8 | RCT | 30 | Continuous aerobic exercise + Resistance training | | No special intervention group |
| Kaur et al. (2022) | India | 24 | RCT | 97 | Nutritional supplements + Cognitive behavioral | | Cognitive behavioral |
| Vizza et al. (2016) | Australia | 12 | RCT | 13 | Resistance training | | No special intervention group |
| Guo et al. (2022) | China | 24 | RCT | 66 | Cognitive behavioral + Short motivational interviews | | Cognitive behavioral |
| Almenning et al. (2015) | Norway | 10 | RCT | 25 | Resistance training | Intermittent aerobic exercise | No special intervention group |
| Pandurevic et al. (2023) | Italy | 16 | RCT | 30 | Ketogenic diet + Mediterranean Diet; | | Mediterranean Diet; |
| Cheshmeh et al. (2022) | Iran | 16 | RCT | 194 | Nutritional supplements + Hypocaloric diet | | Hypocaloric |
| Kazemi et al. (2018) | Canada | 16 | RCT | 61 | High-protein diet | | Low fat diet |
| McBrearity et al. (2020) | Canada | 12 | RCT | 60 | High-protein diet | | Low fat diet |
| Abdollahi et al. (2019) | Iran | 12 | RCT | 74 | Cognitive behavioral | | No special intervention group |
| Karamali et al. (2018) | Iran | 8 | RCT | 60 | High-protein diet | | Low fat diet |
| Li et al. (2022) | China | 12 | RCT | 42 | Tai Chi | | Continuous aerobic exercise |
| Jafari-Sfidvajani et al. (2018) | Iran | 12 | RCT | 54 | Nutritional supplements + Hypocaloric | | Hypocaloric diet |

| Author, year | Country | Lasting days (weeks) | Study type | Sample | Intervention | | Control |
|---|---|---|---|---|---|---|---|
| | | | | | T1 | T2 | |
| *Vigorito et al. (2007)* | Italy | 12 | RCT | 90 | Continuous aerobic exercise | | No special intervention group |
| *Patten et al. (2022)* | Australia | 12 | RCT | 24 | Intermittent aerobic exercise | | Continuous aerobic exercise |
| *Costa et al. (2018)* | Brazil | 16 | RCT | 27 | Continuous aerobic exercise | | No special intervention group |
| *Bruner, Chad & Chizen (2006)* | Canada | 12 | RCT | 12 | Resistance training + Cognitive behavioral | | Cognitive behavioral |
| *Pandit et al. (2022)* | India | 48 | RCT | 48 | Continuous aerobic exercise | | No special intervention group |
| *Benham et al. (2021)* | Canada | 24 | RCT | 47 | Intermittent aerobic exercise | Continuous aerobic exercise | No special intervention group |
| *Nybacka, Hellström & Hirschberg (2017)* | Sweden | 16 | RCT | 57 | Hypocaloric | Cognitive behavioral | Hypocaloric + Cognitive behavioral |
| *Philbois et al. (2022)* | Brazil | 16 | RCT | 75 | Continuous aerobic exercise | Intermittent aerobic exercise | No special intervention group |
| *Asemi et al. (2014)* | Iran | 8 | RCT | 50 | Dietary Approaches to Stop Hypertension | | No special intervention group |
| *Sørensen et al. (2012)* | Denmark | 24 | RCT | 27 | High-protein diet | | No special intervention group |

**Note:**
RCT, randomized controlled trial.

single interventions and multiple interventions combined into a study measure. Table 1 summarizes the basic characteristics of the 29 included articles.

## Quality assessment of the included studies

Of the 29 studies included, 17 randomized controlled trials did not detail the randomization of the selection of subjects for inclusion. One study was assessed as high risk when the selection of subjects was not randomized, but was randomized during the grouping process. Four studies did not detail the randomization grouping procedure, and the rest described the randomization of the grouping. In terms of blinding setup, participants and investigators were not blinded in four studies, and five studies were not blinded to the measurements, so these six studies were judged to be at high risk of bias. None of the studies had missing data, and only eight articles did not give all the data, so they were rated as "unknown risk" because we did not know

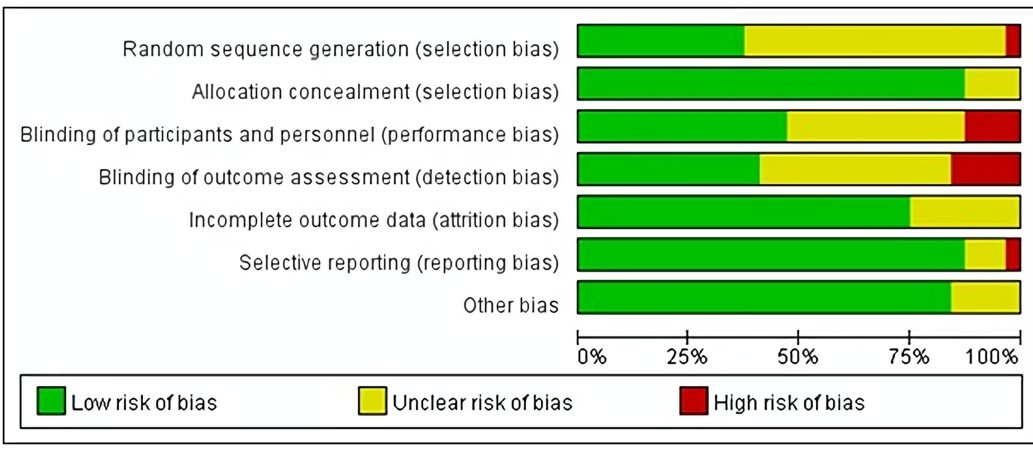

**Figure 2 Assessment figure risk of bias.**               

**Table 2 Results of traditional meta-analysis.**

|  | Comparison | *n* | SMD (95% CI) | *I²* | *P* |
|---|---|---|---|---|---|
| Weight | 18 *vs.* 1 | 4 | −2.81 [−5.05 to −0.57] | NC | <0.05 |
| BMI | 18 *vs.* 1 | 6 | −1.44 [−1.28 to −0.61] | 16% | <0.05 |
|  | 2 *vs.* 3 | 3 | −1.32 [−2.48 to −0.17] | NC | <0.05 |
| WC | 2 *vs.* 3 | 3 | −3.70 [−5.91 to −1.48] | NC | <0.05 |
| WHR | 18 *vs.* 1 | 3 | −0.03 [−0.06 to −0.01] | NC | <0.05 |

**Note:**
1, No special intervention group; 2, Nutritional supplements + Hypocaloric diet; 3, Hypocaloric diet; 18, Continuous aerobic exercise; NC, Not Calculable.

whether the data were complete. The detailed quality assessment results of all studies are shown in Fig. 2.

## Results of traditional meta-analysis

A total of 22 articles were included for traditional paired meta-analysis and the results are shown in Table 2. (1) In terms of weight reduction in patients with PCOS, a total of four studies have evaluated the effect of continuous aerobic exercise on weight loss in patients with PCOS. The results showed that continuous aerobic exercise (SMD = −2.81, 95% CI [−5.05 to −0.57]) was significantly better than the no specific intervention group in terms of weight loss, and the difference was statistically significant. (2) In terms of reducing BMI in patients with PCOS, a total of nine studies evaluated the effectiveness of various modalities in improving BMI in patients with PCOS. The results showed that continuous aerobic exercise (SMD = −1.44, 95% CI [−1.28 to −0.61]) was significantly better than the no specific intervention group with statistically significant difference; nutritional supplements + hypocaloric diet (SMD = −1.32, 95% CI [−2.48 to −0.17]) was significantly better than hypocaloric diet with statistically significant difference. (3) In terms of reducing WC in patients with PCOS, a total of three studies evaluated the effect of nutritional supplements + hypocaloric diet on improving WC in patients with PCOS. The results

showed that nutritional supplements + hypocaloric diet (SMD = −3.70, 95% CI [−5.91 to −1.48]) were significantly better than hypocaloric diet, and the difference was statistically significant. (4) In terms of reducing WHR in patients with PCOS, a total of three studies evaluated the effect of continuous aerobic exercise on improving WHR in patients with PCOS. The results showed that continuous aerobic exercise (SMD = −0.03, 95% CI [−0.06 to −0.01]) was significantly better than that of the group with no specific intervention, and the difference was statistically significant. Overall, continuous aerobic exercise and a diet supplemented with nutrients (*e.g.*, cysteine-rich spinach extract, *Lactobacillus rhamnosus*, vitamin D3) have shown certain advantages in reducing patients' weight levels.

## Results of network meta-analysis

Among the 29 randomized controlled trials in this study, we included a total of four primary outcome indicators (BMI, weight, WC, and WHR), and we will report the results of the NMA in detail. We conducted a NMA of the randomized controlled trials included for each outcome indicator, with each indicator forming a network map that provides direct insight into the direct comparisons or indirect comparisons between different interventions for each indicator, as well as the number of studies that made comparisons. The studies that included the indicator of WC did not form a better closed loop, so we compared them as two closed loops. The specific results are shown in Fig. 3, and the ordinal numbers in the figure correspond to the interventions shown in Table 3.

## Improvement effect of BMI

A total of 17 studies included BMI as an outcome indicator. We included 17 studies in a NMA for inconsistency model testing, which showed $P = 0.167$, and the results of the node-split method showed that all $P > 0.05$, indicating that there was no significant difference between direct and indirect comparisons, and that our analysis had a high degree of consistency and confidence. The results of NMA showed that Taichi (MD = 5.15, 95% CI [0.23–10.06], $P < 0.05$) was more effective in improving weight management in patients with PCOS compared to intermittent aerobic exercise, and other interventions were not statistically different in direct and indirect comparisons (Fig. S1). From the cumulative ranking table, it can be seen that for the outcome indicator of BMI, the top three non-pharmacological interventions among the 14 non-pharmacological interventions were: (1) nutritional supplements + hypocaloric diet (SUCRA = 83.3%). (2) Taichi (SUCRA = 81.5%). (3) nutritional supplements + cognitive behavioral (SUCRA = 74.6%); SUCRA ranking details are shown in Table 4.

## Improvement effect on body weight

A total of 18 studies included weight as an outcome indicator. The results of the inconsistency model showed $P = 0.511$, and the results of the node-split method showed all $P > 0.05$, indicating that there was no significant difference between direct and indirect comparisons, and that our analyses had a high degree of consistency and confidence. NMA showed that high protein diet + continuous aerobic exercise, high protein diet + continuous aerobic exercise + resistance exercise, Taichi, continuous aerobic exercise,

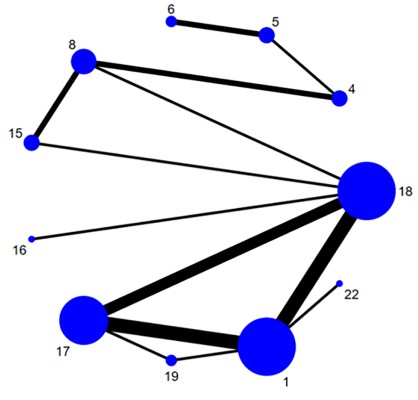

**BMI**

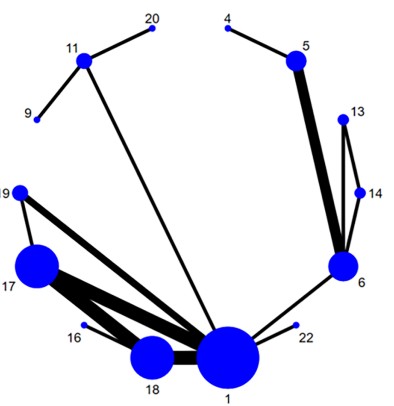

**Weight**

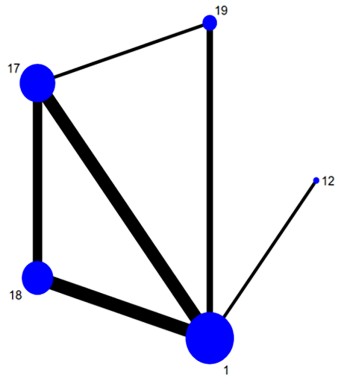

**WC₁**

**WC₂**

| | Note |
| --- | --- |
| BMI | Network map for body mass index |
| Weight | Network map for body weight |
| WC₁ | Network map for waist circumference group 1 |
| WC₂ | Network map for waist circumference group 2 |
| WHR | Network map for waist-hip ratio |

**Figure 3 Network relationship diagram.**

**Table 3 Serial number of the corresponding intervention.**

| Number | Intervening measure | Number | Intervening measure |
| --- | --- | --- | --- |
| 1 | No special intervention group | 12 | Continuous aerobic exercise + Resistance training |
| 2 | Nutritional supplements + Hypocaloric diet | 13 | High-protein diet + Continuous aerobic exercise |
| 3 | Hypocaloric diet | 14 | High-protein diet + Continuous aerobic exercise + Resistance training |
| 4 | Mediterranean diet | 15 | Acupuncture |
| 5 | Low fat diet | 16 | Tai Chi |
| 6 | High-protein diet | 17 | Intermittent aerobic exercise |
| 7 | Nutritional supplements | 18 | Continuous aerobic exercise |
| 8 | Ketogenic diet + Mediterranean Diet | 19 | Resistance training |
| 9 | Nutritional supplements + Cognitive behavioral | 20 | Resistance training + Cognitive behavioral |
| 10 | Cognitive behavioral + Short motivational interviews | 21 | Hypocaloric diet + Cognitive behavioral |
| 11 | Cognitive behavioral | 22 | Dietary Approaches to Stop Hypertension |

**Table 4 SUCRA sorting table.**

| Index | Ranking the top three measures | SUCRA values for the top three interventions (%) |
|---|---|---|
| BMI | ① Nutritional supplements + hypocaloric diet; | 83.3 |
| | ② Tai chi; | 81.5 |
| | ③ Nutritional supplements + cognitive behavioral; | 74.6 |
| Weight | ① Tai chi; | 89.6 |
| | ② High-protein diet+ continuous aerobic exercise; | 79.9 |
| | ③ Continuous aerobic exercise; | 74.6 |
| $WC_1$ | ① Continuous aerobic exercise; | 92.6 |
| | ② Continuous aerobic exercise + resistance training; | 71.5 |
| | ③ No special intervention group; | 47.2 |
| $WC_2$ | ① Nutritional supplements + hypocaloric diet; | 84.8 |
| | ② Nutritional supplements + cognitive behavioral; | 76.5 |
| | ③ Hypocaloric diet; | 59.2 |
| WHR | ① Tai chi; | 79.5 |
| | ② Continuous aerobic exercise; | 69.6 |
| | ③ Intermittent aerobic exercise; | 49.2 |

**Note:**
BMI, body mass index; $WC_1$, waist circumference group 1; $WC_2$, waist circumference group 2; WHR, waist-hip ratio.

intermittent aerobic exercise, resistance exercise and dietary approaches to stop hypertension were more advantageous than nutritional supplementation + cognitive behavioral therapy in reducing body weight of patients with PCOS, $P < 0.05$. For the remaining 24 interventions, there were statistically significant differences between direct and indirect comparisons of weight improvement (Fig. S2). From the cumulative ranking table, it can be seen that for the outcome indicator of body weight, the top three non-pharmacological interventions among the 14 non-pharmacological interventions were: (1) Taichi (SUCRA = 89.6%). (2) High-protein diet + continuous aerobic exercise (SUCRA = 79.9%). (3) Continuous aerobic exercise (SUCRA = 74.6), and the SUCRA ranking is shown in Table 4.

### Improvement effect of waist circumference

A total of 16 studies were included in the NMA of WC, but, unfortunately, the 16 studies did not form a complete closed loop when the network graphs were constructed, so we decided to divide the 16 included studies into two groups for the analysis ($WC_1$, $WC_2$). Inconsistency model test for both studies showed $P_1 = 0.186$ and $P_2 = 0.599$, and the results of node splitting method for both groups showed $P > 0.05$, which indicates that there is no significant difference between direct and indirect comparisons, and that our analyses have a high degree of consistency and credibility. We performed a NMA of the data from each of the two groups, and the results showed that in the $WC_1$ group: resistance exercise (MD = 8.8, 95% CI [0.37–17.23], $P < 0.05$) did not have a better effect on improving WC compared to no specific intervention; The combination intervention of continuous aerobic exercise + resistance exercise (MD = 9.8, 95% CI [0.78–18.82], $P < 0.05$) and continuous

aerobic exercise (MD = −10.05, 95% CI [−19.38 to 0.71], $P < 0.05$) was more advantageous in improving WC compared to intermittent aerobic exercise. In the WC 2 group: four interventions, nutritional supplements + low-calorie diet, low-calorie diet, nutritional supplements + cognitive behavioral therapy, and nutritional supplements + motivational interviewing, were more effective in reducing WC values compared to the combination intervention of low-calorie diet + cognitive behavioral therapy (Figs. S3 and S4). Finally, we compared the data of the two groups in cumulative order, and the results showed that the top three intervention effects in the $WC_1$ group were: (1) Continuous aerobic exercise (SUCRA = 92.6%). (2) Continuous aerobic exercise + resistance training (SUCRA = 71.5%). (3) No special intervention group (SUCRA = 47.2%). The top three intervention effects in the $WC_2$ group were: (1) Nutritional supplements + hypocaloric diet (SUCRA = 84.8%). (2) Nutritional supplements + cognitive behavioral (SUCRA = 76.5%). (3) Hypocaloric (SUCRA = 49.2%), and the ranking of SUCRA is detailed in Table 4.

## Improvement effect of waist-to-hip ratio

A total of four studies were included in the NMA of waist-hip ratio, and the test of inconsistency for the four studies showed $P = 0.473$, and the results of the node splitting method showed all $P > 0.05$, indicating that there was no significant difference between the direct and indirect comparisons, and that there was a high degree of consistency and confidence in our analyses. NMA showed that all direct and indirect comparisons were not statistically significant (Fig. S5). We compared the cumulative rankings of the five interventions included in the four studies, and the results showed that the top three interventions in terms of improving the waist-hip ratio were: (1) Taichi (SUCRA = 79.5%). (2) Continuous aerobic exercise (SUCRA = 69.6%). (3) Intermittent aerobic exercise (SUCRA = 49.2%). The ranking of SUCRA is shown in Table 4.

## Publication bias

As an outcome indicator to test for publication bias, we plotted a funnel plot of BMI in patients with PCOS, as shown in Fig. 4. Based on what is presented in the figure, we concluded that there is a low possibility of publication bias in this study.

## DISCUSSION

In this systematic review and NMA we assessed the effects of non-pharmacological interventions on anthropometric measures (weight, BMI, WC, and WHR) in people with overweight PCOS. The evaluation included data from 29 randomized controlled trials (1,565 patients), involving 22 different combinations or separate interventions, and showed the effects of different interventions for each different index through effect ranking, providing a clinical reference for non-pharmacological interventions to improve weight loss and metabolism inpatients with PCOS. Our study adds to the evidence related to the effectiveness of nonpharmacologic interventions in improving weight loss for women living with PCOS, particularly those experiencing overweight or obesity. The risk of bias was generally low in the included randomized controlled trials, but the sample size was small, and only a few interventions were compared in pairwise comparisons in TMA.

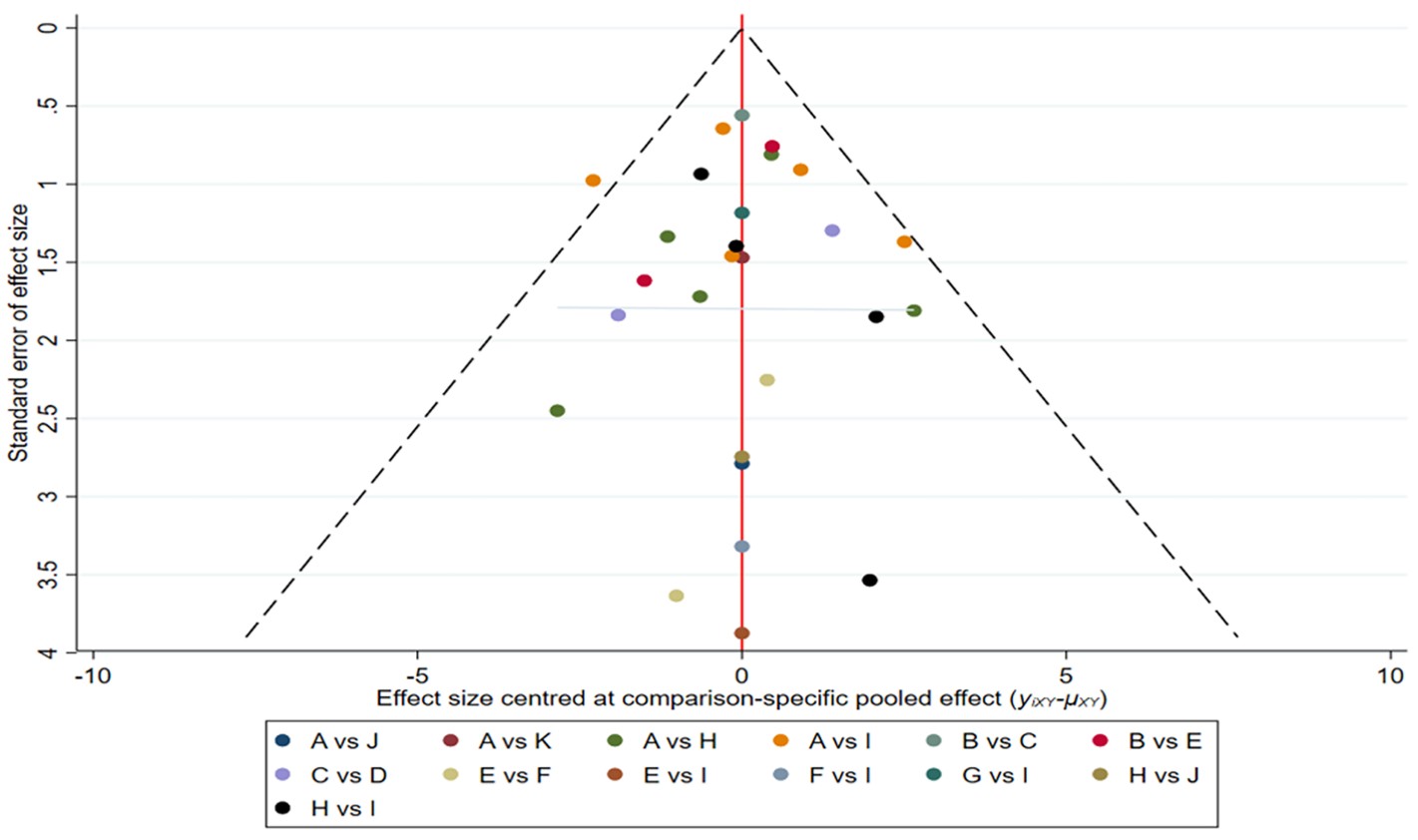

**Figure 4  Funnel plots.**

Some studies suggest that short-term (8–16 weeks) low-calorie dietary interventions may support weight management and improve hyperandrogenism and metabolic dysfunction in individuals with PCOS and elevated BMI (*Deshmukh et al., 2023*). Oral nutritional supplements can also be effective in lowering body weight and BMI (*Li et al., 2021*). The results of the present study, which showed that the nutritional supplement + low-calorie diet combination intervention had the best effect in reducing BMI in patients with PCOS, are similar to the results of a systematic evaluation of nutritional supplements for improving body mass in patients with PCOS by *Li et al. (2021)*. However, *Li et al. (2021)* ranked the effects of the nutritional supplement interventions only and did not unite them to the dietary status of the patients. So, this study is the first to suggest that a combination intervention of nutritional supplements + low-calorie diet may be the more effective in reducing BMI in patients. However, in clinical practice, keeping patients on nutritional supplements is fine, but keeping them on a low-calorie diet is a more difficult task. Notably, Taichi had an unexpected effect in lowering BMI in patients with PCOS, where it came in second. Possible factors for Taichi to improve physical functioning and reduce BMI are that Taichi requires the practitioner to maintain a lower posture for a long period of time, which helps to build the patient's leg strength and strengthens the patient's core. Another factor is that Taichi's exercise is moderately intense and has a high degree of generalizability in countries with a Confucian culture (*Qi et al., 2023*). We divided the

studies included with the outcome indicator of WC into two groups for NMA, which showed that the top-ranked interventions in the $WC_1$ and $WC_2$ groups were: continuous aerobic exercise and nutritional supplements + low-calorie diets, respectively, which once again verified that exercise and diet are two of the most effective ways for the human body to reduce weight. However, in patients with PCOS, prolonged low-calorie diets may put patients at risk for hyponutritional states, and low-calorie diets may need to be accompanied by some high-protein foods in order to ensure daily nutrient intake. However, it is clear that this is difficult for patients with PCOS who do not have knowledge of clinical nutrition, which may be a major reason why low-calorie diets are so difficult to carry out in clinical practice (*Hamdy & Horton, 2011*).

The results of our NMA suggest that Taichi may be the best intervention for reducing the waist-to-hip ratio, closely followed by continuous aerobic exercise and interval aerobic exercise. It is noteworthy that Taichi played a better role in these ending indicators we selected, probably because of the uniqueness of its movement style. Taichi attaches great importance to the relationship between the whole body and the parts, emphasizing that the overall movement depends on the local rotation, and the overall movement takes the local rotation as the axis. The rotation of the hips and the waist can achieve the change of the center of gravity during displacement, so as to achieve the coordination of the limbs and the body movement, so that the whole body can be fully mobilized and each part can be exercised (*Li, Qiu & Tie, 2015*). In addition, Taichi combines rigid and soft, alternating between fast and slow, and when dancing Taichi, the muscles of the whole body can be fully coordinated and move in harmony, which is full of philosophy and fun. Besides, compared with continuous aerobic exercise and intermittent aerobic exercise, Taichi exercise is not limited by the site and equipment conditions, and it is more acceptable to the public.

## Limitations of this study

(1) Only one randomized controlled study on Taichi was included in this review, and the number of studies and the sample size of the study were small, which may have led to biased results. (2) This work did not stratify the intensity and duration of exercise intervention implementation, which could lead to potential bias in the results. (3) Dosages and regimens of nutritional supplements were not fully standardized across studies, which may have influenced study results. (4) The outcome indicators of the literature included in this review are relatively dispersed and unfocused, with a small number of studies corresponding to each outcome indicator. (5) The literature search was limited to Chinese and English articles, which may have missed some high-quality studies reported in other languages. Larger samples of multicenter, high-quality clinical studies are needed to evaluate the efficacy and safety of different modalities for weight loss in patients with PCOS.

## CONCLUSION

In conclusion, optimal strategies to support weight-related health outcomes across four key domains (BMI, weight, WC, and WHR) in individuals with PCOS include: nutritional supplements + hypocaloric diet; Taichi; continuous aerobic exercise and Taichi. This is

consistent with the traditional perception that diet + exercise is the best way to reduce weight. Taichi, however, showed an unexpected effect in this NMA, with the greatest advantage in reducing body weight and waist-to-hip ratio in patients with PCOS, which suggests that it is a common misconception that increasing the intensity of exercise is not the only way to reduce body weight and improve health. However, there was variability in the number of interfaces included in this study, which may have contributed to the instability of the results. Therefore, it is recommended that a multicenter, large-sample randomized controlled trial be conducted in the future to validate the effect.

### Funding

This work was supported by the Natural Science Foundation of Gansu Province (23JRRA0945). The authors declare that they have no competing interests. The funders had no role in study design, data collection and analysis, decision to publish, or preparation of the manuscript.

### Grant Disclosures

The following grant information was disclosed by the authors:
Natural Science Foundation of Gansu Province: 23JRRA0945.

### Competing Interests

The authors declare that they have no competing interests.

### Author Contributions

- Rong Hu conceived and designed the experiments, performed the experiments, analyzed the data, prepared figures and/or tables, authored or reviewed drafts of the article, and approved the final draft.
- Lihong Zhang conceived and designed the experiments, analyzed the data, authored or reviewed drafts of the article, and approved the final draft.
- Jingjing Zhu conceived and designed the experiments, prepared figures and/or tables, and approved the final draft.
- Sihua Zhao conceived and designed the experiments, performed the experiments, analyzed the data, prepared figures and/or tables, and approved the final draft.
- Lixue Yin analyzed the data, authored or reviewed drafts of the article, and approved the final draft.
- Junping Hu performed the experiments, prepared figures and/or tables, and approved the final draft.

### Data Availability

This is a systematic review/meta-analysis.

## Supplemental Information

Supplemental information for this article can be found online at http://dx.doi.org/10.7717/peerj.19238#supplemental-information.

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
