# Peer review of "Weight loss effects of non-pharmacological interventions in women with polycystic ovary syndrome: a systematic review and network meta-analysis"

_PeerJ, doi:10.7717/peerj.19238_

## Round 0.1 · original submission · Major Revisions

Thank you for submitting your network meta-analysis. This research addresses a significant gap in the literature by providing a comprehensive comparison of various interventions for [health condition]. The overall concept and methodology are sound, and your findings have the potential to make a valuable contribution to the field.

However, several issues need to be addressed:
1. Typos and Grammatical Errors:
There are numerous typographical errors and grammatical inconsistencies throughout the manuscript. Please carefully proofread the entire document, paying close attention to the following:

Spacing: Ensure there is a single space after each full stop. (e.g., Line 146)
Capitalization: Avoid unnecessary capitalization within sentences. (e.g., Line 116: "ci" should be "CI"; Line 131: "Crls"), Meta-analysis, Continuous aerobic exercise.
Sentence Structure: Break down long sentences with commas into shorter, more precise sentences using full stops. (e.g., Lines 83, 277, 208-215)
Redundant Words (Line 219):
There are duplicated words in line 219. Please remove the redundancy.

2. Transferability Assumption:
Please clarify what you mean by "transferability assumption" in the context of network meta-analysis. This term is not commonly used in NMA, so a clear explanation is essential for readers to understand your methodology.

3. Intervention Duration (Line 148):
Specifying the typical intervention duration in the included studies, which appears to be 12 weeks, would be helpful. This information is crucial for readers to interpret the reported outcomes and understand the time frame required to achieve those effects.

4. Clarity of SMD Reporting:
When reporting standardized mean differences (SMDs), ensure that it is clear which SMD corresponds to each outcome measure. In several instances (Lines 167-169, 195-198, 230, 233-234), you mention multiple characteristics but provide a list of SMDs without clear correspondence. Please revise these sections to link each SMD to its respective outcome explicitly.

5. "Reticulation Meta-Analysis" (Line 222):
The term "reticulation meta-analysis" is not widely recognized. Please define clearly or consider using a more standard term like "network meta-analysis."

6. Inaccurate Reference (Line 281):
The reference citation on line 281 appears to be incorrect. Please verify the author's name (it is not Hu) and ensure the citation is formatted correctly as "Hu et al."

7. Limitations Section:
A dedicated section discussing the limitations of your analysis is missing. Please include a section that addresses the limitations, including the limited evidence on Tai Chi, as the metanalysis included only one article on Tai Chi with a small size.

8. Figure 3 Clarity:
Improve the clarity of Figure 3, particularly the numbers within the figure. Consider using a larger font or a different visual representation to enhance readability.

Please carefully address these points and resubmit your revised manuscript. We believe that with these revisions, your work will make a strong contribution to the field.

Reviewer 1 ·

Basic reporting

The article addresses non-pharmacological interventions of a common and significant condition (PCOS) through meta-analyses. It will be of interest to the readers.
Abstract is structured and is representative of the study. The title is appropriate and the introduction adequately provides an overview of existing literature. The tables and figures are clear and appropriate.
However, some points are suggested for inclusion in the background, and figure 1. Also, language and grammar check is required at a few places.(annotated in the manuscript).

Experimental design

Methodology is elaborate, well-described, and clear. A detailed explanation of the procedure, outcome measures, quality assessment and statistical analyses are given so that it can be replicated if required.

Validity of the findings

The results are well represented. The raw data is provided as supplementary material. The evidence presented is of good quality and valid.
The discussion compares the findings to other researches in the literature and appropriately highlights the similarities and contrasts. It further mentions the limitations. It is suggested to highlight strengths of the study or what it adds to existing literature.
• The conclusion section needs some modifications (included in the manuscript).

Annotated reviews are not available for download in order to protect the identity of reviewers who chose to remain anonymous.

·

Basic reporting

1) The author has mentioned that they have used a total of 5 databases but they have mentioned only two, kindly mention the rest of three as well.
2) The lines 8-10 fails to explain clear message, may be due to (Improper English sentence) used by the authors. e.g. (to assess the effect size for 95% CI) please rectify it.
3) In the line no. 13 the authors have mentioned four outcome measures but in the line no. 14 they haven't used proper punctuation marks to separate them. please make the correction.

Experimental design

No comment

Validity of the findings

No comment

Additional comments

Few minor changes mentioned in the basic reporting part.

·

Basic reporting

The manuscript is written in clear and professional English, with minimal grammatical errors, making it accessible to researchers within the field.

- The introduction provides an adequate context for the study, with a concise summary of the challenges associated with weight management in patients with PCOS. However, the background could benefit from a more comprehensive discussion of the significance of non-pharmacological interventions compared to standard pharmacologic treatments.
- The references are relevant and up-to-date, including studies from reputable sources. It would be helpful to treat the need to change the terminology of PCOS : some recent researches highlighted the perspective on renaming PCOS for an enhanced pathophysiological understanding. The authors could add some references of this aspect.
- The manuscript adheres to PeerJ standards.
- Figures are relevant and well-constructed. However, they could include more descriptive captions to provide context without requiring extensive cross-referencing. Ensure all tables and figures align consistently with the text for clarity.
- The study fits well within the scope of the journal, providing an original contribution by comparing non-pharmacologic interventions for weight loss in PCOS patients.
- The research question is well-defined and meaningful, focusing on identifying effective interventions to improve weight management in PCOS patients.
- The use of SUCRA for ranking interventions is statistically sound and aligns with best practices in network meta-analysis. However, the robustness of the data could be enhanced by discussing the heterogeneity and limitations of the included studies more thoroughly.
- The conclusions are well-supported by the data and directly address the original research question. The authors emphasize that Tai Chi and nutritional supplements combined with a hypocaloric diet are effective strategies, highlighting the practical implications for clinical settings.

Experimental design

no comment

Validity of the findings

no comment

Additional comments

- Add more details on the importance of mediterranean diet to improve this condition and weight loss.
- Add some updated references aspect in renaming PCOS for a better understanding of its pathophysiology.
- Also there some other recent references explaining the role of androgens in clinical aspects and insulinresistance that could enhance the understanding.

·

Basic reporting

This systematic review and network meta-analysis evaluates the effectiveness of non-pharmacological interventions on weight loss management in women with polycystic ovary syndrome (PCOS). English used throughout the manuscript is technically correct, clear and unambiguous. The Literature review organization is good with recent references, and appropriately referenced. A search of five databases, were done, identified randomized controlled trials assessing various non-pharmacological interventions. The results were analyzed using traditional and network meta-analyses and calculate SUCRA ranking scores to determine the most effective interventions. A total of 29 studies, involving 1,565 patients and 22 distinct interventions, were included in the final analysis. Suggestions for improvement : (S.NO:10 SUCRA 13: BMI, (expand and add few words on it) 14: Taijiquan; and tai chi. Add a brief note) All the tables and figures are clear and labelled appropriately a suggestion in Table 3 intervening measure (i-Capital letter)

Experimental design

The study employed a methodology, adhering to the PRISMA guidelines. The design ensured a comprehensive comparison of interventions for overweight patients with PCOS. Objectives are well written following the SMART Acronym research fills in identified knowledge gap.

Validity of the findings

The network meta-analysis demonstrated strong internal validity, supported by the inclusion of a substantial number of studies and patients. This study addresses a significant gap in the literature by systematically comparing non-pharmacological interventions for weight loss in patients with PCOS. The analysis is comprehensive and methodology is well organized and systematic , the heterogeneity in study designs and patient populations necessitates caution in interpreting the conclusions. Authors have mentioned it.

Additional comments

The findings provide valuable insights into non-pharmacological strategies for weight management in women with PCOS, emphasizing the interprofessional holistic approach of integrating nutritional supplements with dietary and exercise interventions. The inclusion of Tai Chi considering it an important alternative, low-intensity physical activities for weight loss and overall health improvement.

---

## Round 0.2 · Minor Revisions

Thank you for your patience and for your thorough revisions to your manuscript. We appreciate the effort you have put into addressing the points raised.
While we see significant improvement and appreciate your responsiveness to the feedback, there are still some grammatical issues and unclear sentences that require further attention before we can accept the manuscript for publication. Specifically, we have noted several instances where the language could be more precise and polished.
• We encourage you to use person-first language when referring to individuals with obesity or “infertility patients". This approach prioritizes the person's humanity and avoids defining them solely by their medical condition. For example, instead of "obese patients," please use "patients with obesity," "individuals with obesity," or "people living with obesity" For example, please carefully review the manuscript and make the necessary edits: Line 47, Line 50, Line 53, Line 55, Line 58, Lines 64, 65, and 68, for example.
• There are still some grammatical and editing errors. For example, I suggest starting the sentence with inclusion criteria included: ----- . Same for exclusion rtieria on lines 85 and 96
• Line 111: Please clarify the nomenclature used here. Specifically, define what "intervention within normal limits" means in this context. Also, please clearly explain the difference between a "blank control" and a "placebo" control. This distinction is important for understanding your methodology.
• Line 140: Please clarify what you mean by "metabolic environment" in the context of PCOS. Provide a more specific definition or explanation.
• Line 164: The phrase "more studies" appears to be a typo. Please correct it to "most studies" or a similar phrase.
• Line 196: You mention WHR (waist-hip Ratio). Please ensure this metric is defined earlier in the manuscript, preferably in the Methods section.
• Line 291: Please clarify what you mean by "complemented the effects." Be more specific about how the effects were complemented.
• Lines 308-313: This appears to be a long and complex sentence. Please consider breaking it down into shorter, more manageable sentences for improved readability.
• Figure 1: The numbers of excluded articles in the different categories do not sum to the total number of excluded articles (114). Please check the numbers and ensure consistency.
• Figure 3 and all other tables and graphs: Please add annotations explaining the abbreviations WHR and WC (Waist Circumference) directly within each table and graph. All tables and graphs should be standalone and understandable without referring to the main text


We are confident that you can address these remaining issues.

Reviewer 1 ·

Basic reporting

The revised version incorporates the suggested changes

Experimental design

The methodology is elaborate, well-described, and clear. A detailed explanation of the procedure, outcome measures, and statistical analyses are given so that it can be replicated if required. As suggested the inclusion section is modified, clear now.

Validity of the findings

Figures have been modified as suggested

Additional comments

The revised version incorporates the suggested changes

·

Basic reporting

I have reviewed the
Author's response letter,
The tracked changes manuscript,
Main manuscript.
The principal investigator effectively responded to all the reviewers comments and provided justifications. This meets the journal's criteria. The manuscript is suitable for publication.

Experimental design

The principal investigator has addressed all the reviewers comments and provided justifications.

Validity of the findings

The principal investigator has addressed all the reviewers comments and provided justifications.

Additional comments

The principal investigator effectively responded to all the reviewers comments and provided justifications. This meets the journal's criteria. The manuscript is suitable for publication.

---

## Round 0.3 · Minor Revisions

Thank you for your revisions. I appreciate the changes you have made; however, I would like to emphasize the need for greater attention to detail in formatting and grammatical accuracy. When using track changes, please ensure that you carefully proofread the document after accepting the edits before submitting it.

There are still some issues with capitalization that need to be addressed. Additionally, I have a remaining comment regarding the wording of "blank" and "convention." In your description of the studies, you state that there is no intervention. However, both "blank" and "convention" are categorized as no intervention, so it is unclear what distinguishes them. The meaning of "placebo" is clear, as it refers to a sugar pill, but what is the specific difference between "blank" and "convention"? Was this terminology used in the original studies? Please clarify this distinction.

LIne 46: delete "e,g"
Line 67: delete " overweight"
Lines 111 and 117: no need to capitalize the first letter of the words
Line 300: (43) is not put after et al.
Line 351: no capitalization in optimal
Line 352" there is a space between : and nutritional
Table 4: there is a comma after some values; no need to capitalize the first letter of the words

I look forward to your revised submission.

---

## Round 0.4 · accepted · Accept

Thank you for your revised submission of the manuscript. We appreciate your prompt response and thorough revision. The authors have addressed all the comments. After reviewing your responses and the revised manuscript, I am pleased to inform you that your article is ready for publication in its current form.